# Diagnosis and treatment of digestive cancers during COVID-19 in Japan: A Cancer Registry-based Study on the Impact of COVID-19 on Cancer Care in Osaka (CanReCO)

**Mari Kajiwara Saito**[ID]*, **Toshitaka Morishima**[ID], **Chaochen Ma, Shihoko Koyama, Isao Miyashiro**

Cancer Control Center, Osaka International Cancer Institute, Osaka City, Osaka Prefecture, Japan

* mari.saito@oici.jp

## Abstract

**Data Availability Statement:** All relevant data are within the paper and its Supporting Information files.

### Background

The coronavirus disease 2019 (COVID-19) affected cancer care in Japan, but the detailed impact on cancer diagnosis and treatment is not well-understood. We aimed to assess the impact of COVID-19 on digestive cancer care in Osaka Prefecture, which has a population of 8.8 million.

### Methods

We conducted a multi-center cohort study, using hospital-based cancer registry (HBCR) data linked to administrative data from 66 designated cancer care hospitals in Osaka. Records of patients diagnosed with cancer of the stomach, colorectum, esophagus, liver, gallbladder or pancreas were extracted from the HBCR data. Baseline characteristics, such as the number of diagnoses, routes to diagnosis and clinical stage, were compared between patients diagnosed in 2019 and those in 2020. We also compared treatment patterns such as the number of treatments (operations, endoscopic surgeries, chemotherapies, radiotherapies), pathological stage and time to treatment for each digestive cancer.

### Results

In total, 62,609 eligible records were identified. The number of diagnoses decreased in 2020, ranging from -1.9% for pancreatic cancer to -12.7% for stomach cancer. Screen-detected cases decreased in stomach and colorectal cancer. The percentage of clinical stage III slightly increased across different cancers, although it was only significant for colorectal cancer. Among 52,741 records analyzed for treatment patterns, the relative decrease in radiotherapy was larger than for other treatments. The median time from diagnosis to operation was shortened by 2–5 days, which coincided with the decrease in operations.

**Funding:** This project is supported by the Osaka Cancer Prevention Fund (Kendai No. 2181) from the Osaka Prefectural Government, Japan. URL: https://www.pref.osaka.lg.jp/kenkozukuri/gankikin_2/index.html MKS is supported by JSPS Grant-in-Aid for Early-Career Scientists (JSPS KAKENHI Grant Number JP22K17340). URL: https://www.jsps.go.jp/j-grantsinaid/03_keikaku/index.html The funders had no role in study design, data collection and analysis, decision to publish, or preparation of the manuscript.

**Competing interests:** The authors have declared that no competing interests exist.

**Abbreviations:** CanReCO, A Cancer Registry-based Study on the Impact of COVID-19 on Cancer Care in Osaka; COVID-19, Coronavirus disease 2019; DCCH, Designated cancer care hospital; DPC, Diagnosis procedure combination; HBCR, Hospital-based cancer registry; IQR, Interquartile range; SED, State of emergency declaration; UICC, Union for International Cancer Control.

## Conclusion

The impact of COVID-19 on cancer care in 2020 was relatively mild compared with other countries but was apparent in Osaka. Further investigation is needed to determine the most affected populations.

## Introduction

The coronavirus disease 2019 (COVID-19) pandemic has disrupted healthcare across the globe, and Japan was no exception. It became necessary to re-allocate and re-organize healthcare resources to cope with COVID-19 in addition to maintaining clinical services for other diseases. When the World Health Organization (WHO) declared the pandemic in March 2020 [1], the first wave of COVID-19 in Japan, although small, had started [2, 3]. Subsequently, aiming to curb the spread of the disease and mitigate disruption of the healthcare system, a state of emergency declaration (SED) was enforced from early April to late May 2020 across the seven most populated prefectures and expanded nationwide in Japan [4]. The SED was, however, less restrictive than lockdowns in other countries; the government requested people to refrain from travelling across prefectures and endorsed social distancing and remote working, but no penalties were set for non-compliance.

Regarding cancer, there may have been healthcare service-induced reduction or delay in providing cancer care [5]. For instance, the Japan Gastroenterological Endoscopy Society recommended suspension of non-urgent endoscopic diagnostics and treatments, due to fear of aerosol infections [6]. Elective operations for cancer were canceled or postponed; a recent report suggests that an estimated 30% of the elective cancer operations in the upcoming 12-week period were canceled in March 2020 in Japan [7].

Also, apprehension about COVID-19 may have led to a patient-induced reduction or delay in accessing cancer care services [5]. Some people might have been concerned that frequent visits to healthcare facilities could increase their risk of infection. This concern would be more common and stronger among cancer patients, as pre-existing comorbidities, including cancer, were reported to be associated with the severity of COVID-19 [8] or increased all-cause mortality [9]. A survey from the United States has reported a substantial concern among cancer patients about contracting COVID-19 by using healthcare facilities [10].

While every effort was made to mitigate the risk of COVID-19 infection and preserve other services, the impact on cancer diagnosis and treatment in Japan of these potentially reduced capacities in healthcare and patients' reluctance to access services, as yet, ill-defined. A Cancer Registry-based Study on the Impact of COVID-19 on Cancer Care in Osaka (CanReCO) project has been launched to investigate the impact of COVID-19 on cancer care and outcomes. In this paper, we aimed to outline the baseline characteristics and treatment patterns of patients diagnosed with six digestive cancers before (2019) and during (2020) the COVID-19 pandemic in Osaka Prefecture, as a part of the CanReCO project. Osaka, the third populous prefecture with a population of 8.8 million, was one of the most affected areas in Japan. It had a total of 65 COVID-19 deaths per million at the end of 2020, the second-highest among 47 prefectures nationwide then [2].

## Methods

### Data sources

We carried out a multi-center cohort study in Osaka Prefecture, using hospital-based cancer registry (HBCR) data linked with administrative data in the CanReCO project. The Osaka

International Cancer Institute (OICI), collaborating with the Council for Coordination of Designated Cancer Care Hospitals in Osaka, requested anonymized records in HBCR and Diagnosis Procedure Combination (DPC) data from 66 out of 67 designated cancer care hospitals (DCCHs) across Osaka Prefecture [11]. The HBCR data was linked to the DPC data at each hospital. The DPC data is one of the most frequently used hospital administrative data in Japan [12]. The CanReCO project includes detailed clinical information in the linked DPC data, but here we aimed to profile patients' characteristics and treatment patterns recorded in the HBCR data. The DCCHs must fulfil certain requirements in resources and patient volume to be accredited as cancer centers [13]. In Osaka Prefecture, 80–90% of cancer operation is covered by DCCHs [14]. However, the OICI is the only cancer-specialized center in Osaka, and the rest are general hospitals providing other services as well as cancer care. Japan has universal healthcare coverage but has no strong gate-keeping system. Thus, people can access these hospitals through referral or directly if they pay an additional fee [15].

In the HBCR data, 99 pieces of information are recorded routinely. Patient information includes age at diagnosis, sex and date of birth. Information on cancer diagnosis and treatment includes the cancer site coded by the International Classification of Diseases for Oncology third edition (ICD-O-3), cancer stage defined by the Union for International Cancer Control (UICC 8th edition) (both clinical and pathological) and date of diagnosis. Date of diagnosis is defined as the date of the earliest and most reliable diagnostic test performed before initiating the first treatment, regardless of whether it was performed in the DCCH or other clinics/ hospitals. In terms of treatment, the HBCR records whether a patient received any of the available surgeries (open, laparoscopic surgery and endoscopic surgery for primary lesion), chemotherapy (including targeted therapy) or radiotherapy (including X-rays, gamma-rays, proton and heavy-particle therapy for primary lesion) and the date of treatment initiation for each procedure. Other treatments, including locoregional therapy for liver cancer (radiofrequency ablation and transarterial chemoembolization), are also recorded in the HBCR. Besides these active treatments, non-invasive treatment, such as monitoring only or palliative care, is recorded. Information on routes to diagnosis (e.g., through screening) and routes to hospital (e.g., through referral) is also available. In the HBCR data, if a patient has multiple tumors in the same site (e.g., two tumors in the ascending and descending colon), they will have multiple records within the same identification number. Identification numbers are assigned within each hospital, thus, if a patient visits multiple hospitals, they would be identified as a different person each time in the HBCR data.

Ethics approval was obtained from the Institutional Review Board at the Osaka International Cancer Institute (approved number 21065). Patient consent was waived by the Board because data for the CanReCO project has been collected for health policy planning and research use. For research use of the data, the Board approved an opt-out approach using a written consent form for each DCCH.

**Study population.** We extracted records of patients diagnosed with primary cancer in the following six sites in the digestive organs: esophagus (ICD-O-3: C15), stomach (C16), colorectum (C18–20), liver and intrahepatic bile ducts (C22), gallbladder or other and unspecified parts of biliary tract (C23–24), or pancreas (C25), from the HBCR data in the CanReCO project. We included records of patients diagnosed with any of the six cancers who were of any age during 2019–2020, and at any stage including carcinoma *in situ*. Records of patients who visited a hospital only to seek a second opinion were excluded. When describing treatment patterns, we further excluded records from hospitals not responsible for planning or initiating a treatment to avoid double counting a patient who visited two or more hospitals for diagnosis or treatment (see above).

**Outcomes and statistical analysis.** We aimed to describe diagnosis and treatment characteristics before and during the COVID-19 pandemic. We compared baseline characteristics such as the total number of diagnoses, age at diagnosis, sex and stage distribution, routes to diagnosis (including screen-detected or through medical check-ups, found while following up other comorbidities or with some symptoms/ autopsy) and routes to hospital (including direct access to a DCCH without referral, through referral, established patient in the hospital) between 2019 and 2020. Japan has both population-based and opportunistic screening programs for stomach cancer using fluoroscopy or gastroscopy for those aged 50 or over [16], and for colorectal cancer using fecal immunochemical tests for those aged 40 or over [17]. Endoscopies and ultrasounds can be offered by gastroenterological specialists in some clinics at the primary care level. We also compared the number of treatments by year for operation (including both open and laparoscopic surgery), endoscopic surgery, chemotherapy, radiotherapy, other treatment (including locoregional therapy for liver cancer only) and non-invasive treatment (diagnosis, monitoring or palliative care only). Time to treatment was defined as the interval between the date of diagnosis and the date of initiating a treatment. The chi-square test or Wilcoxon rank-sum test were used where appropriate. In addition to the yearly comparison, we examined monthly figures for the number of treatments and time to treatment to see more detailed changes over time. Stata 16 MP (StataCorp, College Station, Texas, US) was used for all analyses.

## Results

The CanReCO project identified a total of 171,480 records for patients diagnosed with cancer in all cancer sites during 2019 and 2020 (**Fig 1** and **S1 Fig**). Of these, a total of 63,985 records for six cancer sites of the digestive organs were found. After excluding 1376 records which were second opinions, we had 62,609 records for the description of baseline characteristics.

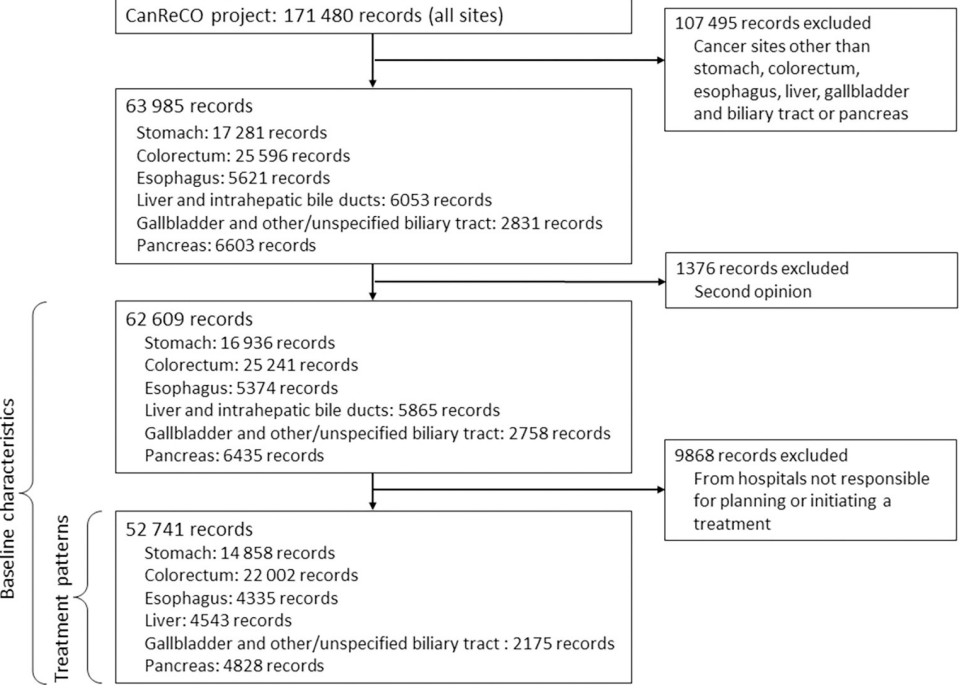

**Fig 1. Flow chart of eligible records with six digestive cancers in the CanReCO project.**

We further excluded 9868 records from hospitals not responsible for planning or initiating a treatment when describing treatment patterns. Finally, our description of treatment patterns included 52,741 records.

## Baseline characteristics of newly diagnosed records

A yearly comparison of the baseline characteristics is shown in **Tables 1 and 2**. The scale of the change in the number of diagnoses varied by cancer site in the digestive organs, as in other sites (**S2 Fig**). A substantial decline in the total number of diagnoses was seen in cancer of the stomach (relative change -12.7%) and esophagus (-10.4%), in contrast to cancer of the colorectum (-5.6%) and pancreas (-1.9%). The median age at diagnosis shifted by +1 year for all six cancers, and the fall in the number of diagnoses was relatively large among younger age groups. Sex distribution changed little for all digestive cancers. Altered routes to diagnosis were evident only in cancer of the stomach and colorectum, but the relative decline was most profound in screen-detected cases among all routes for all digestive cancers. There was strong evidence that routes to hospital changed in cancer of the stomach, colorectum and liver, with a notable relative decline in the proportion of referred patients in liver cancer. Although the number of diagnoses decreased in all cancers, there was no evidence that stage distribution changed, except for colorectal cancer. Monthly figures showed that the drop in the number of diagnoses was considerable in April and May for all six cancers except pancreatic cancer (**S1 Table**).

## Treatment patterns and time to treatment

The change in the total number of treatments was similar to that for diagnoses (**Tables 3 and 4**). The largest decline in the number of treatments was seen in stomach cancer (-13.1%), while there was minimal change in pancreatic cancer (-0.2%). The number of operations, endoscopic surgeries and chemotherapies followed similar patterns to the total number of treatments. However, in all cancers except stomach cancer, the scale of relative decline for radiotherapies was more extensive than that for overall treatment. A notable change was seen in the number of operations in esophageal cancer, with a relative +4.1% increase.

The median time from diagnosis to operation was shortened by 2–5 days, except for pancreatic cancer (+15 days). The median time to endoscopic surgery was also shortened by 2–3 days in 2020 for stomach and esophageal cancer, but the figure remained at 0 for colorectal cancer as the date of endoscopic surgery was defined as the date of diagnosis following the guidelines which recommend avoiding biopsy [18]. When we investigated median time to first treatment of any kind, it was shortened by two days also for pancreatic cancer in 2020 (**S3 Fig** and **S2 Table**).

A more detailed analysis of change by month highlighted a substantial dip in the number of operations during the first wave of COVID-19 (March to May 2020) in the following four cancers: stomach (-39.6%), colorectal (-27.0%), esophageal (-34.9%) and liver cancer (-28.8%) (**Fig 2** and **S3 Table**). After July 2020, the number of operations recovered in colorectal and esophageal cancer, while a small drop persisted in stomach cancer. The number of operations among pancreaticobiliary cancers was hardly affected during April and May 2020, but a relatively sharp decline (> -20%) was seen in July and August 2020. As the number of operations dropped, there was generally a clear shortening trend in time to operation except for pancreatic cancer. Similarly, the number of endoscopic surgeries and time to treatment decreased during the first wave; the largest reduction in the number of endoscopic surgeries was seen in esophageal cancer: -48.1% in May 2020 (**Fig 3** and **S4 Table**).

**Table 1. Baseline characteristics of patients diagnosed with six digestive cancers in the CanReCO project.**

| Cancer site | Stomach | | | Colorectum | | | Esophagus | | |
|---|---|---|---|---|---|---|---|---|---|
| Year of diagnosis | 2019 | 2020 | Relative change | 2019 | 2020 | Relative change | 2019 | 2020 | Relative change |
| Total number of diagnoses | 9042 | 7894 | -12.7% | 12,985 | 12,256 | -5.6% | 2835 | 2539 | -10.4% |
| Median age at diagnosis | 74 | 75 | p[b]<0.001 | 71 | 72 | p[b]<0.001 | 71 | 72 | p[b] = 0.008 |
| IQR | 68–80 | 68–80 | | 64–78 | 64–79 | | 65–77 | 65–78 | |
| Age group, n (%) | | | | | | | | | |
| 0–54 | 531 (5.9) | 444 (5.6) | -16.4% | 1447 (11.1) | 1321 (10.8) | -8.7% | 176 (6.2) | 161 (6.3) | -8.5% |
| 55–64 | 1046 (11.6) | 827 (10.5) | -20.9% | 1916 (14.8) | 1884 (15.4) | -1.7% | 525 (18.5) | 427 (16.8) | -18.7% |
| 65–74 | 3097 (34.3) | 2629 (33.3) | -15.1% | 4443 (34.2) | 3925 (32.0) | -11.7% | 1095 (38.6) | 970 (38.2) | -11.4% |
| 75–84 | 3406 (37.7) | 3021 (38.3) | -11.3% | 3978 (30.6) | 3865 (31.5) | -2.8% | 884 (31.2) | 840 (33.1) | -5.0% |
| 85– | 962 (10.6) | 973 (12.3) | +1.1% | 1201 (9.3) | 1261 (10.3) | +5.0% | 155 (5.5) | 141 (5.6) | -9.0% |
| Female, n (%) | 2816 (31.1) | 2545 (32.2) | p[c] = 0.126 | 5353 (41.2) | 5205 (42.5) | p[c] = 0.045 | 541 (19.1) | 518 (20.4) | p[c] = 0.225 |
| Routes to diagnosis, n (%) | | | p[d]<0.001 | | | p[d] = 0.015 | | | p[d] = 0.130 |
| Screen-detected or medical check-ups | 1206 (13.9) | 906 (12.0) | -24.9% | 2299 (18.8) | 2016 (17.4) | -12.3% | 317 (11.7) | 245 (10.1) | -22.7% |
| Found while f/u of other comorbidities | 3362 (38.8) | 3095 (41.0) | -7.9% | 3784 (30.9) | 3699 (31.9) | -2.2% | 973 (36.0) | 914 (37.7) | -6.1% |
| Symptomatic presentation or autopsy[a] | 4092 (47.3) | 3554 (47.0) | -13.1% | 6160 (50.3) | 5873 (50.7) | -4.7% | 1414 (52.3) | 1267 (52.2) | -10.4% |
| Unknown | (382 [4.2]) | (339 [4.3]) | | (742 [5.7]) | (668 [5.5]) | | (131 [4.6]) | (113 [4.5]) | |
| Routes to hospital, n (%) | | | p[d]<0.001 | | | p[d]<0.001 | | | p[d] = 0.325 |
| No referral | 784 (8.7) | 557 (7.1) | -29.0% | 1345 (10.4) | 1102 (9.0) | -18.1% | 160 (5.7) | 132 (5.2) | -17.5% |
| Referral | 5821 (64.4) | 5036 (63.8) | -13.5% | 8801 (67.8) | 8260 (67.4) | -6.1% | 1943 (68.6) | 1699 (67.0) | -12.6% |
| Established patient | 2227 (24.6) | 2095 (26.6) | -5.9% | 2582 (19.9) | 2575 (21.0) | -0.3% | 699 (24.7) | 670 (26.4) | -4.1% |
| Others | 208 (2.3) | 201 (2.6) | -3.4% | 253 (2.0) | 315 (2.6) | +24.5% | 31 (1.1) | 35 (1.4) | +12.9% |
| Unknown | (2) | (5) | | (4) | (4) | | (2) | (3) | |
| Clinical stage (UICC), n (%) | | | p[d] = 0.558 | | | p[d] = 0.004 | | | p[d] = 0.308 |
| 0 | 1 (<0.1) | 0 (0.0) | NA | 2065 (19.8) | 1876 (19.2) | -9.2% | 350 (14.0) | 318 (13.9) | -9.1% |
| I | 5029 (62.3) | 4298 (61.2) | -14.5% | 2549 (24.5) | 2226 (22.7) | -12.7% | 965 (38.5) | 861 (37.6) | -10.8% |
| II | 693 (8.6) | 599 (8.5) | -13.6% | 1822 (17.5) | 1731 (17.7) | -5.0% | 283 (11.3) | 256 (11.2) | -9.5% |
| III | 863 (10.7) | 780 (11.1) | -9.6% | 2264 (21.8) | 2281 (23.3) | +0.8% | 361 (14.4) | 381 (16.6) | +5.5% |
| IV | 1493 (18.5) | 1343 (19.1) | -10.0% | 1707 (16.4) | 1684 (17.2) | -1.3% | 545 (21.8) | 473 (20.7) | -13.2% |
| NA | (8) | (6) | | (10) | (11) | | (133 [4.7]) | (113 [4.5]) | |
| Unknown | (955 [10.6]) | (868 [11.0]) | | (2568 [19.8]) | (2447 [20.0]) | | (198 [7.0]) | (137 [5.4]) | |

Abbreviations: f/u, follow-up; IQR, interquartile range; NA, not applicable; UICC, Union for International Cancer Control. (a) Records with autopsy were 1 for each year in stomach cancer. (b) P-values of the Wilcoxon rank-sum test. (c) P-values of the $\chi^2$ test. (d) P-values of the $\chi^2$ test excluding unknown and NA status in each variable. Note that the percentages in square brackets are omitted for NA and Unknown status for $n \leq 15$ for routes to diagnosis, routes to hospital and clinical stage. The denominators of the percentages for routes to diagnosis except unknown status, routes to hospital except unknown status and clinical stage 0 to IV are the total number of records with unknown and NA status removed in each variable. The denominators of the percentages for unknown routes to diagnosis or NA/unknown clinical stage are the total number of records.

**Table 2. Baseline characteristics of patients diagnosed with six digestive cancers in the CanReCO project.**

| Cancer site | Liver | | | Gallbladder | | | Pancreas | | |
|---|---|---|---|---|---|---|---|---|---|
| Year of diagnosis | 2019 | 2020 | Relative change | 2019 | 2020 | Relative change | 2019 | 2020 | Relative change |
| Total number of diagnoses | 3084 | 2781 | -9.8% | 1435 | 1323 | -7.8% | 3249 | 3186 | -1.9% |
| Median age at diagnosis | 75 | 76 | $p^b$ = 0.363 | 76 | 77 | $p^b$ = 0.082 | 73 | 74 | $p^b$ = 0.002 |
| IQR | 68–81 | 69–81 | | 69–82 | 70–83 | | 66–79 | 67–80 | |
| Age group, n (%) | | | | | | | | | |
| 0–54 | 175 (5.7) | 164 (5.9) | -6.3% | 64 (4.5) | 56 (4.2) | -12.5% | 252 (7.8) | 229 (7.2) | -9.1% |
| 55–64 | 337 (10.9) | 301 (10.8) | -10.7% | 131 (9.1) | 112 (8.5) | -14.5% | 440 (13.5) | 419 (13.2) | -4.8% |
| 65–74 | 956 (31.0) | 832 (29.9) | -13.0% | 437 (30.5) | 373 (28.2) | -14.6% | 1102 (33.9) | 1000 (31.4) | -9.3% |
| 75–84 | 1216 (39.4) | 1123 (40.4) | -7.6% | 556 (38.8) | 542 (41.0) | -2.5% | 1135 (34.9) | 1166 (36.6) | +2.7% |
| 85– | 400 (13.0) | 361 (13.0) | -9.8% | 247 (17.2) | 240 (18.1) | -2.8% | 320 (9.9) | 372 (11.7) | +16.3% |
| Female, n (%) | 917 (29.7) | 846 (30.4) | $p^c$ = 0.567 | 645 (45.0) | 600 (45.4) | $p^c$ = 0.831 | 1473 (45.3) | 1519 (47.7) | $p^c$ = 0.060 |
| Routes to diagnosis, n (%) | | | $p^d$ = 0.189 | | | $p^d$ = 0.693 | | | $p^d$ = 0.182 |
| Screen-detected or medical check-ups | 122 (4.3) | 88 (3.4) | -27.9% | 55 (4.0) | 43 (3.4) | -21.8% | 160 (5.2) | 129 (4.3) | -19.4% |
| Found while f/u of other comorbidities | 1911 (67.7) | 1779 (69.1) | -6.9% | 480 (35.3) | 437 (35.0) | -9.0% | 1045 (34.1) | 1027 (33.9) | -1.7% |
| Symptomatic presentation or autopsy[a] | 791 (28.0) | 709 (27.5) | -10.4% | 826 (60.7) | 770 (61.6) | -6.8% | 1857 (60.7) | 1875 (61.9) | 1.0% |
| Unknown | (260 [8.4]) | (205 [7.4]) | | (74 [5.2]) | (73 [5.5]) | | (187 [5.8]) | (155 [4.9]) | |
| Routes to hospital, n (%) | | | $p^d$<0.001 | | | $p^d$ = 0.343 | | | $p^d$ = 0.434 |
| No referral | 127 (4.1) | 105 (3.8) | -17.3% | 135 (9.4) | 102 (7.7) | -24.4% | 236 (7.3) | 230 (7.2) | -2.5% |
| Referral | 1900 (61.6) | 1542 (55.5) | -18.8% | 1021 (71.2) | 941 (71.2) | -7.8% | 2419 (74.5) | 2333 (73.3) | -3.6% |
| Established patient | 985 (32.0) | 1058 (38.1) | +7.4% | 244 (17.0) | 246 (18.6) | +0.8% | 539 (16.6) | 552 (17.3) | +2.4% |
| Others | 71 (2.3) | 73 (2.6) | +2.8% | 34 (2.4) | 33 (2.5) | -2.9% | 54 (1.7) | 68 (2.1) | +25.9% |
| Unknown | (1) | (3) | | (1) | (1) | | (1) | (3) | |
| Clinical stage (UICC), n (%) | | | $p^d$ = 0.082 | | | $p^d$ = 0.183 | | | $p^d$ = 0.684 |
| 0 | 0 (0.0) | 1 (<0.1) | NA | 20 (2.1) | 8 (0.9) | -60.0% | 36 (1.3) | 32 (1.1) | -11.1% |
| I | 1193 (47.2) | 1147 (49.5) | -3.9% | 158 (16.5) | 167 (18.7) | +5.7% | 770 (27.4) | 793 (28.3) | +3.0% |
| II | 594 (23.5) | 472 (20.4) | -20.5% | 202 (21.0) | 175 (19.6) | -13.4% | 344 (12.2) | 334 (11.9) | -2.9% |
| III | 404 (16.0) | 373 (16.1) | -7.7% | 234 (24.4) | 213 (23.9) | -9.0% | 374 (13.3) | 400 (14.3) | +7.0% |
| IV | 336 (13.3) | 325 (14.0) | -3.3% | 346 (36.0) | 328 (36.8) | -5.2% | 1287 (45.8) | 1245 (44.4) | -3.3% |
| NA | (85 [2.8]) | (83 [3.0]) | | (7) | (5) | | (3) | (3) | |
| Unknown | (472 [15.3]) | (380 [13.7]) | | (468 [32.6]) | (427 [32.3]) | | (435 [13.4]) | (379 [11.9]) | |

Abbreviations: f/u, follow-up; IQR, interquartile range; NA, not applicable; UICC, Union for International Cancer Control. (a) Records with autopsy were 1 in 2019 in pancreatic cancer only. (b) P-values of the Wilcoxon rank-sum test. (c) P-values of the $\chi^2$ test. (d) P-values of the $\chi^2$ test excluding unknown and NA status in each variable. Note that the percentages in square brackets are omitted for NA/unknown status for n ≤ 15 for routes to diagnosis, routes to hospital and clinical stage. The denominators of the percentages for routes to diagnosis except unknown status, routes to hospital except unknown status and clinical stage 0 to IV are the total number of records with unknown and NA status removed in each variable. The denominators of the percentages for unknown routes to diagnosis or NA/unknown clinical stage are the total number of records.

**Table 3. Treatment patterns and pathological stage in patients with six digestive cancers in the CanReCO project.**

| Cancer site | Stomach | | | Colorectum | | | Esophagus | | |
|---|---|---|---|---|---|---|---|---|---|
| Year of diagnosis | 2019 | 2020 | Relative change | 2019 | 2020 | Relative change | 2019 | 2020 | Relative change |
| Total number of treatments | 7950 | 6908 | -13.1% | 11,321 | 10,681 | -5.7% | 2267 | 2068 | -8.8% |
| Median age at diagnosis | 74 | 75 | $p^c$<0.001 | 72 | 72 | $p^c$<0.001 | 71 | 72 | $p^c$ = 0.016 |
| IQR | 68–80 | 69–80 | | 65–79 | 64–79 | | 65–77 | 65–78 | |
| Female, n (%) | 2442 (30.7) | 2217 (32.1) | $p^d$ = 0.071 | 4619 (40.8) | 4498 (42.1) | $p^d$ = 0.048 | 418 (18.4) | 423 (20.5) | $p^d$ = 0.094 |
| Number of operations | 3215 | 2763 | -14.1% | 6642 | 6209 | -6.5% | 556 | 579 | +4.1% |
| Number of endoscopic surgeries | 3372 | 2923 | -13.3% | 3985 | 3763 | -5.6% | 923 | 831 | -10.0% |
| Number of chemotherapies | 1839 | 1519 | -17.4% | 2824 | 2601 | -7.9% | 906 | 828 | -8.6% |
| Number of radiotherapies | 59 | 67 | +13.6% | 223 | 203 | -9.0% | 469 | 414 | -11.7% |
| Number of non-invasive treatments only[a] | 718 | 704 | -1.9% | 526 | 511 | -2.9% | 216 | 194 | -10.2% |
| Median time from diagnosis to operation[b] | 34 days | 29 days | $p^c$<0.001 | 27 days | 23 days | $p^c$<0.001 | 71 days | 69 days | $p^c$ = 0.312 |
| IQR (days) | 21–57 | 18–49 | -14.7%[e] | 15–42 | 13–37 | -14.8%[e] | 48–90 | 43–90 | -2.8%[e] |
| Median time from diagnosis to endoscopic surgery[b] | 35 days | 32 days | $p^c$ = 0.001 | 0 days | 0 days | $p^c$ = 0.012 | 38 days | 36 days | $p^c$ = 0.055 |
| IQR (days) | 20–52 | 19–49 | -8.6%[e] | 0–19 | 0–15 | 0.0%[e] | 24–56 | 22–52 | -5.3%[e] |
| Pathological stage (UICC), n (%) | | | $p^f$ = 0.130 | | | $p^f$ = 0.810 | | | $p^f$ = 0.825 |
| 0 | 0 (0.0) | 0 (0.0) | 0.0% | 3099 (31.3) | 2916 (31.4) | -5.9% | 268 (24.5) | 242 (24.2) | -9.7% |
| I | 4757 (77.7) | 4073 (76.2) | -14.4% | 2245 (22.7) | 2080 (22.4) | -7.3% | 719 (65.7) | 666 (66.5) | -7.4% |
| II | 567 (9.3) | 496 (9.3) | -12.5% | 1902 (19.2) | 1839 (19.8) | -3.3% | 48 (4.4) | 42 (4.2) | -12.5% |
| III | 575 (9.4) | 563 (10.5) | -2.1% | 1841 (18.6) | 1693 (18.2) | -8.0% | 38 (3.5) | 38 (3.8) | 0.0% |
| IV | 223 (3.6) | 215 (4.0) | -3.6% | 803 (8.1) | 772 (8.3) | -3.9% | 21 (1.9) | 13 (1.3) | -38.1% |
| Post-neoadjuvant therapy | (226 [2.8]) | (162 [2.4]) | | (343 [3.0]) | (287 [2.7]) | | (356 [15.7]) | (371 [17.9]) | |
| No operation/endoscopic surgery | (1567 [19.7]) | (1375 [19.9]) | | (1050 [9.3]) | (1055 [9.9]) | | (771 [34.0]) | (658 [31.8]) | |
| NA/Unknown | (5/30 [0.4]) | (2/22 [0.3]) | | (3/35 [0.3]) | (9/30 [0.3]) | | (44 [1.9]/2) | (34 [1.6]/4) | |

Abbreviations: IQR, interquartile range; NA, not applicable; UICC, Union for International Cancer Control. (a) Non-invasive treatments include diagnosis, monitoring or palliative care only. (b) Date of operation was missing in 1 patient in esophageal cancer, thus excluded. Date of endoscopic surgery was missing in 10 patients in stomach and 3 in colorectal cancer, thus excluded. (c) P-values of the Wilcoxon rank-sum test. (d) P-values of the $\chi^2$ test. (e) Relative change of the point estimates of median time to treatment. (f) P-values of the $\chi^2$ test for pathological stage 0 to IV only. Note that the percentages in square brackets are omitted for NA/unknown stage with n ≤ 15 for pathological stage. The denominators of the percentages for stage 0 to IV are the total number of records with pathological stage restricted to 0 to IV. The denominators of the percentages for post-neoadjuvant therapy, no operation/endoscopic surgery and NA/unknown stage are the total number of records with all stages.

## Discussion

This study outlined the descriptive statistics of the impact of COVID-19 on cancer diagnosis and treatment in Osaka Prefecture, comparing HBCR data for the whole of 2019 and 2020. The number of diagnoses fell by more than 10% in stomach and esophageal cancer, while pancreatic cancer was hardly affected by COVID-19. The timing of the fall in the monthly number of diagnoses coincided with the COVID-19 waves. A relative drop in screen-detected cases was largest among all routes to diagnosis in all six digestive cancers. The level of decline in the

**Table 4. Treatment patterns and pathological stage in patients with six digestive cancers in the CanReCO project.**

| Cancer site | Liver | | | Gallbladder | | | Pancreas | | |
|---|---|---|---|---|---|---|---|---|---|
| Year of diagnosis | 2019 | 2020 | Relative change | 2019 | 2020 | Relative change | 2019 | 2020 | Relative change |
| Total number of treatments | 2353 | 2190 | -6.9% | 1142 | 1033 | -9.5% | 2417 | 2411 | -0.2% |
| Median age at diagnosis | 75 | 76 | $p^c$ = 0.026 | 77 | 77 | $p^c$ = 0.029 | 73 | 74 | $p^c$ = 0.790 |
| IQR | 68–81 | 69–81 | | 70–82 | 71–83 | | 67–79 | 67–80 | |
| Female, n (%) | 700 (29.8) | 662 (30.2) | $p^d$ = 0.073 | 501 (43.9) | 457 (44.2) | $p^d$ = 0.862 | 1092 (45.2) | 1157 (48.0) | $p^d$ = 0.050 |
| Number of operations | 764 | 724 | -5.2% | 555 | 499 | -10.1% | 806 | 798 | -1.0% |
| Number of chemotherapies | 921 | 853 | -7.4% | 392 | 359 | -8.4% | 1478 | 1498 | +1.4% |
| Number of radiotherapies | 79 | 68 | -13.9% | 21 | 15 | -28.6% | 171 | 143 | -16.4% |
| Number of other treatments[a] | 997 | 907 | -9.0% | | | | | | |
| Number of non-invasive treatments only[b] | 399 | 347 | -13.0% | 336 | 304 | -9.5% | 601 | 621 | +3.3% |
| Median time from diagnosis to operation | 40 days | 36 days | $p^c$ = 0.012 | 32 days | 28 days | $p^c$ = 0.094 | 41 days | 56 days | $p^c$ = 0.002 |
| IQR (days) | 27–57 | 26–52 | -10.0%[e] | 11–48 | 13–45 | -12.5%[e] | 23–84 | 26–93 | +36.6%[e] |
| Pathological stage, n (%) | | | $p^f$ = 0.189 | | | $p^f$ = 0.931 | | | $p^f$ = 0.824 |
| 0 | 0 (0.0) | 2 (0.3) | - | 46 (9.4) | 44 (9.9) | -4.3% | 71 (12.0) | 66 (14.1) | -7.0% |
| I | 337 (48.7) | 351 (53.5) | +4.2% | 100 (20.4) | 98 (22.0) | -2.0% | 222 (37.6) | 169 (36.1) | -23.9% |
| II | 257 (37.1) | 218 (33.2) | -15.2% | 190 (38.7) | 171 (38.4) | -10.0% | 197 (33.4) | 155 (33.1) | -21.3% |
| III | 84 (12.1) | 69 (10.5) | -17.9% | 115 (23.4) | 101 (22.7) | -12.2% | 63 (10.7) | 53 (11.3) | -15.9% |
| IV | 14 (2.0) | 16 (2.4) | +14.3% | 40 (8.2) | 31 (7.0) | -22.5% | 37 (6.3) | 25 (5.3) | -32.4% |
| Post-neoadjuvant therapy | (56 [2.4]) | (58 [2.7]) | | (20 [1.8]) | (17 [1.7]) | | (209 [8.7]) | (329 [13.7]) | |
| No operation | (1542 [65.5]) | (1438 [65.7]) | | (575 [50.4]) | (521 [50.4]) | | (1605 [66.4]) | (1609 [66.7]) | |
| NA/Unknown | (59 [2.5]/4) | (37 [1.7]/1) | | (7/49 [4.3]) | (6/44 [4.3]) | | (9/4) | (4/1) | |

Abbreviations: IQR, interquartile range; NA, not applicable; UICC, Union for International Cancer Control. (a) Other treatments include locoregional therapies such as radiofrequency ablation and transarterial chemoembolization. (b) Non-invasive treatments include diagnosis, monitoring or palliative care only. (c) P-values of the Wilcoxon rank-sum test. (d) P-values of the $\chi^2$ test. (e) Relative change of the point estimates of median time to treatment. (f) P-values of the $\chi^2$ test for pathological stage 0 to IV only. Note that the percentages in square brackets are omitted for NA/unknown stage with n ≤ 15 for pathological stage. The denominators of the percentages for stage 0 to IV are the total number of records with pathological stage restricted to 0 to IV. The denominators of the percentages for post-neoadjuvant therapy, no operation and NA/unknown stage are the total number of records with all stages.

number of treatments was almost analogous to that in the number of diagnoses for all cancers, but the relative decline in radiotherapies was generally more apparent than that in other treatments. We also showed that the median time to treatment was shortened alongside the decrease in the number of treatments in most cancers on a monthly basis.

When comparing cancer sites, there were several interesting points about their change. Although endoscopy is necessary for the diagnosis of stomach, colorectal and esophageal cancers, the drop in number of diagnoses was more profound in cancers of the upper digestive tract (> -10%) than colorectal cancer (-5.6%). A potential explanation for this difference in the level of change might be akin to the view held in Australia [19], that colonoscopy is indicated and necessary if a patient tested positive for a fecal immunochemical test, whereas there is no such prerequisite for gastroscopy. A survey for endoscopic services in Japan reported a shortage in personal protective equipment [20], which may have led to limiting endoscopic services in general during the pandemic. However, aerosol infection through the respiratory tract might have been more avoided than fecal shedding.

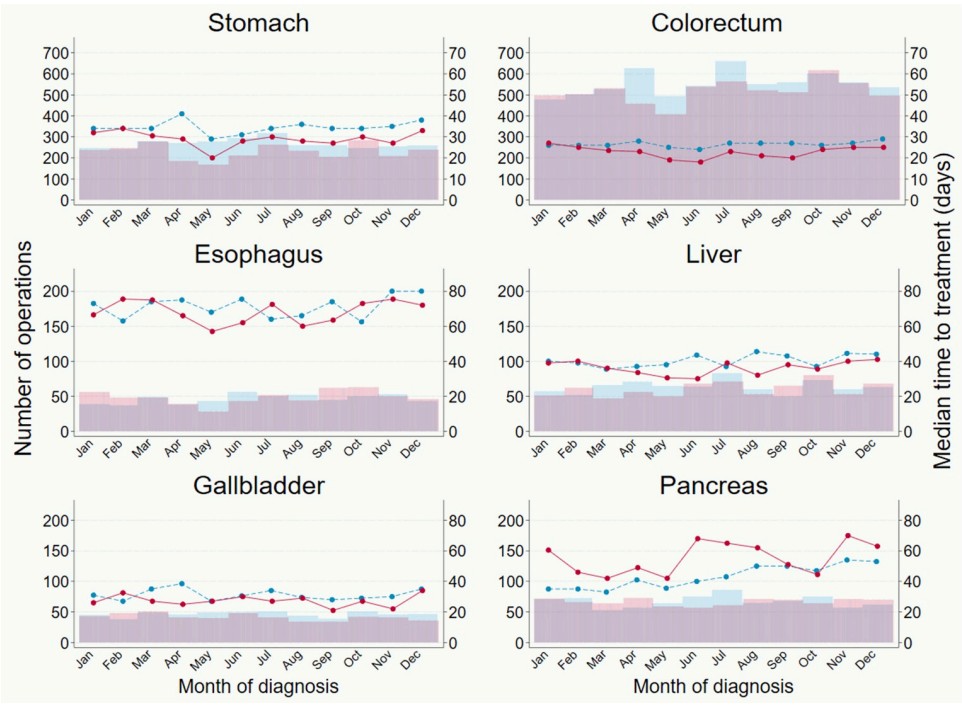

**Fig 2. Number of operations and median time to operation for six digestive cancers.** Bar graphs indicate number of operations (left axis: blue 2019, pink 2020) and line graphs indicate monthly median time in days from diagnosis to operation (right axis: blue dash 2019, red solid 2020). Note that both the right and left axes are different for stomach and colorectal cancer.

The drop in the number of liver cancer cases diagnosed (-9.8%) was larger than the average of all sites combined (-6.9%). The considerable decrease in liver cancer diagnoses might be because around 70% of patients with hepatocellular carcinoma, which constitutes over 90% of all liver cancers in Japan [21], have liver cirrhosis [22]. This comorbidity reflects a generally poor systemic condition with an immunocompromised status. It can be fatal if patients are infected with COVID-19: hazard ratios of COVID-19-related death increased around twofold compared to patients with no chronic liver disease [23, 24]. Along with a relatively large fall in the number of referrals to DCCHs among this group, the decline in diagnoses might have been induced by both healthcare professionals and patients; doctors in nearby clinics and patients might have suspended regular follow-ups by ultrasound and tumor markers. The fall in the number of gallbladder cancer cases was noticeable among younger age groups. In contrast to liver cancer, this population is unlikely to have equally severe underlying conditions but might have suspended attending imaging examinations.

Among six digestive cancers, the number of diagnoses was relatively preserved for pancreatic cancer. Considering that this group usually presents with some symptoms and no recommended screening exists, the observed minor impact of COVID-19 on the number of diagnoses is as we expected.

Regarding routes to diagnosis, a report from the Japan Cancer Society suggested that screening uptake and the number of cases detected through screening in 2020 dropped from 2019 levels by around 30% and 20% for stomach and colorectal cancer, respectively, in Japan [25, 26]. The decline in the screening uptake seemed to be more substantial (20–30%) among older people (age $\geq$ 60) than younger people (around 10%) [25]. In this study, we showed that the relative decline in diagnoses was substantial in the younger age groups in these two cancers

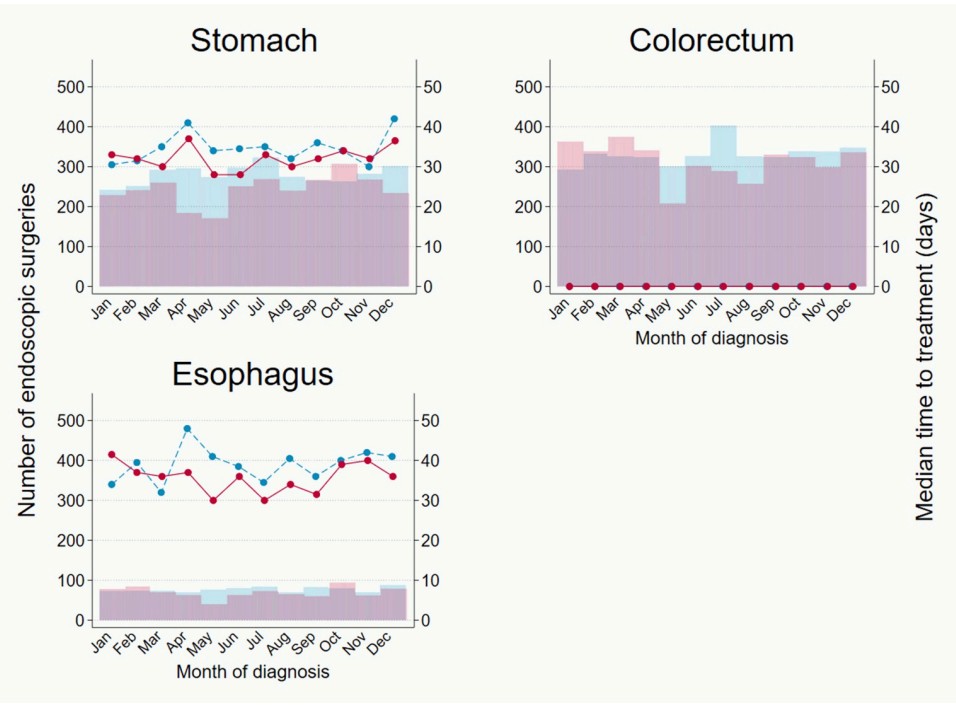

**Fig 3. Number of endoscopic surgeries and median time to endoscopic surgery for stomach, colorectal and esophageal cancer.** Bar graphs indicate number of endoscopic surgeries (left axis: blue 2019, pink 2020) and line graphs indicate monthly median time in days from diagnosis to endoscopic surgery (right axis: blue dash 2019, red solid 2020).

and the absolute number of declines was largest in the 65–74 age group. Although no upper age limit is set for screening programs in Japan, we observed no reduction in the proportion of records detected through screening among those aged 85 and over (**S5 Table**). Our results imply that the patient-induced reduction might have been considerable among asymptomatic patients from age groups under 75 years.

We observed no evidence of a shift towards higher clinical or pathological stages in this study population for most cancers. A change in clinical stage was evident only for colorectal cancer, but this might be due to the large population size and other cancers showed similar trends toward a slight increase in stage III. Also, for esophageal cancer, reduced endoscopic surgeries and increased operations may together imply that some patients with stage 0 may have shifted to higher stages. However, these results should be interpreted with great caution because we do not know the extent to which the disease in the missed cases had progressed due to late detection. A delay to diagnosis or treatment can occur for the following reasons: time from the onset of a symptom to contacting a healthcare professional, time from the initial presentation to diagnosis, or time from referral to a specialist care took longer than anticipated [27]. Delayed diagnosis or treatment might be associated with an increased mortality in some cancers [27, 28]. In a study from England, it was estimated that diagnostic delays can lead to a 6–17% increase in avoidable death up to 5 years from diagnosis for colorectal and esophageal cancer [29].

Regarding treatment, a relative reduction in radiotherapy was more obvious than in other treatments in all cancers except stomach cancer. For esophageal and pancreatic cancer in particular, radiotherapy can be either curative or palliative [30, 31], but the number of patients with a pathological stage at post-neoadjuvant therapy increased for both cancers. The

declining radiotherapy treatment in two cancer sites may reflect the desire to avoid contracting COVID-19 through a lengthy stay or frequent visits to hospital for palliative radiotherapy. For rectal cancer as well, radiotherapy can be used for either purpose [32]. However, as shown in a relatively small population with pathological stage at post-neoadjuvant therapy in colorectal cancer, radical radiotherapy is not as common in Japan as in Western countries.

Although the real consequences of the COVID-19 pandemic are yet to be known, we consider that measuring its impact on cancer care uncovers the stability, strengths and weaknesses of the healthcare system in a country. In addition to our results, understanding patients' experiences during the pandemic and examining both short- and long-term cancer survival may provide insights into which point of care can be improved to achieve better cancer outcomes. At this point, we assume the impact of the pandemic on cancer care in our study population during 2020 was modest compared with other high-income countries [33–35]. A potential explanation is that the burden of the pandemic during 2020 was not so high in Japan as in other countries. The cumulative number of confirmed cases and deaths was around 236,000 and 3400, respectively, across Japan at the end of 2020 [36]. The national figures correspond to 1870 total cases and 27.7 total deaths per million, not reaching 10% of those in other high-income countries [36]. Another explanation could be related to the country's healthcare system. Although Japan has a publicly funded insurance system [15], healthcare is mainly provided by private institutions with strong professional autonomy. The government subsidized hospitals that accepted COVID-19 patients [37], but not all hospitals might have accepted COVID-19 patients during 2020; only 17% (81 out of 476 hospitals, 436 are private hospitals) in Osaka Prefecture accepted COVID-19 patients as of December 2020 [38]. Role-sharing between hospitals might have occurred during the initial phase of the pandemic. Thus, cancer care might have been less disrupted than we initially believed.

Of note, our study showed that the interval from diagnosis to treatment was shortened during the pandemic. The shortened interval after the pandemic reflects underlying overstretched cancer care services due to a growth in the number of patients before the COVID-19 pandemic in Japan. Considering a drop in both diagnoses and treatments in 2020, a backlog can be anticipated for years to come. Overloading in the cancer care services may lead to a treatment delay when the missed cases are absorbed. Proper allocation of resources should be planned to prepare for the foreseen backlog to avoid further delay. The situation of the pandemic has been continuously changing with the evolution of variant strains and the development of treatment and vaccines [39, 40]. The COVID-19 pandemic continues across Japan, and the CanReCO project will continue to monitor cancer care and its outcome. The full impact of COVID-19 on cancer will be revealed only after a few years.

The strength of our study is that we used data specific to Osaka, the third most populous prefecture in Japan and one of those most severely hit by COVID-19 [2]. A report of stage distribution from two hospitals in Kanagawa Prefecture exists [41], but our study could be valuable due to its scale and details encompassing nearly all DCCHs across Osaka Prefecture. The national average decrease in the number of diagnoses in 2020 (all sites combined, reference 2019) was -4.6% [42, 43], while we observed a -6.9% reduction in Osaka Prefecture (**S1 Fig**). We also showed monthly changes in time to treatment together with the number of treatments, which would confer strengths to our study evaluating the overall effect of COVID-19 on cancer care. We could effectively contrast the change in diagnoses and treatments by investigating several cancers together in the digestive organs, which have distinctive features in terms of their etiologies and presentations.

We also recognize several limitations, mostly due to the nature of the HBCR data. Firstly, as a reference, only one year (2019) was available for the pre-COVID period. Therefore, we could not account for any yearly trend in the number of diagnoses and treatments. However, we

were able to capture statistics for the whole two-year period and also to compare 2019 and 2020 in detail by month in the CanReCO project. Secondly, the HBCR data is not population-based. Some patients have multiple primaries in the same organ in our dataset. Thus, the number of diagnoses in the population-based data could be lower than our figures as it is estimated to only be two-thirds of the HBCR number [44]. In addition, not all patients attend DCCHs, but some may be diagnosed or treated in non-designated hospitals. These patients will be not recorded in the HBCR in our project. However, because 80–90% of the patients undergo operations in a DCCH in Osaka Prefecture [14], we assume that the percentage of patients receiving treatment of any kind in a DCCH could be even higher. A concern about selection bias in this study is that patients diagnosed in 2020 might have changed where they received cancer care. In particular, the most vulnerable population might have chosen more local, non-designated hospitals after the SED. In that case, our results may overestimate the level of decline due to COVID-19 in 2020. Another issue is that some patients might have attended more than one hospital for cancer care because the HBCR data regards these patients as different people. However, we eliminated this problem by restricting the records from hospitals responsible for planning or initiating a treatment when investigating the treatment patterns. Our results on the relative change in the number of operations were comparable to the estimates derived from the National Clinical Database encompassing all surgical procedures in Japan [45]; therefore, we assume our results are valid. Nonetheless, our results may not be generalizable to other countries or other areas in Japan. It is because the scale and timing of the pandemic, healthcare system, and the baseline characteristics of a population, including socioeconomic situations, health-seeking behaviors and screening uptakes, differ originally between and within countries. Thirdly, we can only make inferences but cannot conclude which factors, healthcare service- or patient-induced or both, affected cancer care. To date, there has been no formal analysis, nationwide or at the prefectural level, regarding how much the diagnostic intensity, such as imaging or endoscopy activities, was depleted in Japan. The final limitation is that information on elective or emergency operation, socioeconomic status, comorbidities and measures of performance status has been not linked to these results. Identifying the affected populations and assessing the detailed situation is a key to proper reallocation of resources and aligning supply with patients' needs. More detailed data on these characteristics that can be derived from DPC data are now available and will be able to reveal which populations were most affected by COVID-19.

In conclusion, we captured a moderately reduced number of diagnoses and treatments of digestive cancers following the COVID-19 in 2020 in Osaka Prefecture, but the scale of change varied by cancer site. Time to treatment was shortened after the start of COVID-19, potentially reflecting overstretched cancer care before the pandemic, thus preparation is needed for the anticipated backlog. We will investigate further impacts on cancer care and outcomes and identify affected populations in forthcoming CanReCO project reports

## Supporting information

**S1 Fig. Number of diagnoses (all sites) in the CanReCO project and number of COVID-19 cases detected, Osaka, Japan, 2019 and 2020.**
(PDF)

**S2 Fig. Number of diagnoses in 2019 and relative change in 2020 (reference: 2019) by cancer site in the CanReCO project, Osaka, Japan.**
(PDF)

**S3 Fig. Number of treatments and median time from diagnosis to first treatment by month of diagnosis for six digestive cancers in the CanReCO project, Osaka, Japan, 2019 and 2020.**
(PDF)

**S1 Table. Number of diagnoses and clinical stage 0 & I, percentage of clinical stage 0 & I and relative change by month for six digestive cancers, Osaka, Japan, 2019 and 2020.**
(PDF)

**S2 Table. Median time from diagnosis to first treatment for six digestive cancers in the CanReCO project, Osaka, Japan, 2019 and 2020.**
(PDF)

**S3 Table. Number of operations and median time to operation by month for six digestive cancers in the CanReCO project, Osaka, Japan, 2019 and 2020.**
(PDF)

**S4 Table. Number of endoscopic surgeries and median time to endoscopic surgery by month for three digestive cancers in the CanReCO project, Osaka, Japan, 2019 and 2020.**
(PDF)

**S5 Table. Number of screen-detected cases by age group for stomach and colorectal cancer in the CanReCO project, Osaka, Japan, 2019 and 2020.**
(PDF)

## Acknowledgments

We would like to thank 66 designated cancer care hospitals for participation and data provision.

We would like to thank Dr Julia Mortimer for English language proofreading.

## Author Contributions

**Conceptualization:** Mari Kajiwara Saito, Toshitaka Morishima, Chaochen Ma, Shihoko Koyama, Isao Miyashiro.

**Data curation:** Mari Kajiwara Saito, Toshitaka Morishima, Chaochen Ma, Shihoko Koyama.

**Formal analysis:** Mari Kajiwara Saito.

**Funding acquisition:** Isao Miyashiro.

**Investigation:** Toshitaka Morishima, Isao Miyashiro.

**Methodology:** Mari Kajiwara Saito, Toshitaka Morishima, Chaochen Ma, Shihoko Koyama, Isao Miyashiro.

**Project administration:** Toshitaka Morishima, Isao Miyashiro.

**Resources:** Toshitaka Morishima, Isao Miyashiro.

**Supervision:** Toshitaka Morishima, Chaochen Ma, Shihoko Koyama, Isao Miyashiro.

**Validation:** Mari Kajiwara Saito, Toshitaka Morishima, Shihoko Koyama.

**Visualization:** Mari Kajiwara Saito.

**Writing – original draft:** Mari Kajiwara Saito.

**Writing – review & editing:** Mari Kajiwara Saito, Toshitaka Morishima, Chaochen Ma, Shihoko Koyama, Isao Miyashiro.

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
