## [Decision Letter · Decision Letter 0]

12 Aug 2022

PONE-D-22-19182Diagnosis and treatment of digestive cancers during COVID-19 in Japan: A cancer registry-based study on the impact of COVID-19 on cancer care in Osaka (CanReCO)PLOS ONE

Dear Dr. Kajiwara Saito,

Thank you for submitting your manuscript to PLOS ONE. After careful consideration, we feel that it has merit but does not fully meet PLOS ONE’s publication criteria as it currently stands. Therefore, we invite you to submit a revised version of the manuscript that addresses the points raised during the review process.

We look forward to receiving your revised manuscript.

Kind regards,

Ahmet Murt

Academic Editor

PLOS ONE

Journal Requirements:

Additional Editor Comments:

The paper needs some revisions and if you believe that you can do them we will be expecting the revised version of your manuscript. Reviewer comments are attached.

Reviewers' comments:

Reviewer's Responses to Questions

**Comments to the Author**

1. Is the manuscript technically sound, and do the data support the conclusions?

Reviewer #1: Yes

Reviewer #2: Yes

2. Has the statistical analysis been performed appropriately and rigorously? 

Reviewer #1: Yes

Reviewer #2: Yes

3. Have the authors made all data underlying the findings in their manuscript fully available?

Reviewer #1: Yes

Reviewer #2: Yes

4. Is the manuscript presented in an intelligible fashion and written in standard English?

Reviewer #1: No

Reviewer #2: Yes

5. Review Comments to the Author

Reviewer #1: Dear Editor, thank you so much for inviting me to revise this manuscript.

This study addresses a current topic.

The manuscript is quite well written and organized. English could be improved.

Figures and tables are comprehensive and clear.

The introduction explains in a clear and coherent manner the background of this study.

We suggest the following modifications:

• Introduction section: although the authors correctly included important papers in this setting, we believe a couple of studies should be cited within the introduction ( PMID: 35109688; PMID: 32658591 ), only for a matter of consistency. We think it might be useful to introduce the topic of this interesting study.

• Methods and Statistical Analysis: nothing to add.

• Discussion section: Very interesting and timely discussion. Of note, the authors should expand the Discussion section, including a more personal perspective to reflect on. For example, they could answer the following questions – in order to facilitate the understanding of this complex topic to readers: what potential does this study hold? What are the knowledge gaps and how do researchers tackle them? How do you see this area unfolding in the next 5 years? We think it would be extremely interesting for the readers.

However, we think the authors should be acknowledged for their work. In fact, they correctly addressed an important topic, the methods sound good and their discussion is well balanced.

One additional little flaw: the authors could better explain the limitations of their work, in the last part of the Discussion.

We believe this article is suitable for publication in the journal although major revisions are needed. The main strengths of this paper are that it addresses an interesting and very timely question and provides a clear answer, with some limitations.

We suggest a linguistic revision and the addition of some references for a matter of consistency. Moreover, the authors should better clarify some points.

Reviewer #2: The authors analyzed GI cancer diagnosis and treatment patterns in Osaka, Japan, comparing the COVID pandemic and the time before. Since the impact of the COVID pandemic is different according to the medical environment of each country, the data in this paper will help compare the situation of each country.

6. PLOS authors have the option to publish the peer review history of their article (what does this mean?). If published, this will include your full peer review and any attached files.

Reviewer #1: No

Reviewer #2: No

---

## [Author Response · Author response to Decision Letter 0]

25 Aug 2022

Journal Requirements

Thank you for your guidance. We have amended the manuscript to meet PLOS ONE’s style requirements.

Thank you for providing the guidance. We have included additional details in the main text as follows (Line 82-85), and added the same text to the “Ethics Statement” field of the submission form.

“Patient consent was waived by the Board because data for the CanReCO project has been collected for health policy planning and research use. For research use of the data, the Board approved an opt-out approach using a written consent form for each DCCH.”

Thank you for the guidance. We have updated our Data Availability Statement as follows.

“All relevant data are within the manuscript and its Supporting Information files.”

We have added an additional Supporting Information file to describe the manuscript (S5 Table), and all relevant data are now provided within the paper. As stated in the previous section, Data Availability Statement was changed accordingly.

A section on “Separate caption for each figure” was added after the References and before the section on Supporting Information. 

Thank you for your guidance. We have reviewed our reference list. We confirm that it is complete and correct, and we do not include papers that have been retracted. Please note that the following citations were added to the revised manuscript.

39. World Health Organization. Therapeutics and COVID-19: Living guideline, 14 July 2022. Geneva: 2022 (WHO/2019-nCoV/therapeutics/2022.4). Licence: CC BY-NC-SA 3.0 IGO.

40. Altmann DM, Boyton RJ. COVID-19 vaccination: The road ahead. Science. 2022;375(6585):1127-32. doi: 10.1126/science.abn1755.

43. Okuyama A, Watabe M, Makoshi R, Takahashi H, Tsukada Y, Higashi T. Impact of the COVID-19 pandemic on the diagnosis of cancer in Japan: analysis of hospital-based cancer registries. Jpn J Clin Oncol. 2022. doi: 10.1093/jjco/hyac129.

 

Response to reviewers

5. Review Comments to the Author

Reviewer #1: Dear Editor, thank you so much for inviting me to revise this manuscript.

This study addresses a current topic.

The manuscript is quite well written and organized. English could be improved.

Figures and tables are comprehensive and clear.

The introduction explains in a clear and coherent manner the background of this study.

Thank you for taking the time to review our manuscript. We appreciate your constructive feedback on the manuscript.

We suggest the following modifications:

• Introduction section: although the authors correctly included important papers in this setting, we believe a couple of studies should be cited within the introduction ( PMID: 35109688; PMID: 32658591 ), only for a matter of consistency. We think it might be useful to introduce the topic of this interesting study.

Thank you for your comment. We have added a phrase on the topic and cited relevant articles in the Discussion section (Line 319-320) as follows.

“Proper allocation of resources should be planned to prepare for the foreseen backlog to avoid further delay. The situation of the pandemic has been continuously changing with the evolution of variant strains and the development of treatment and vaccines [39, 40]. The COVID-19 pandemic continues across Japan, and the CanReCO project will continue to monitor cancer care and its outcome. The full impact of COVID-19 on cancer will be revealed only after a few years.”

• Methods and Statistical Analysis: nothing to add.

Thank you for your comment.

• Discussion section: Very interesting and timely discussion. Of note, the authors should expand the Discussion section, including a more personal perspective to reflect on. For example, they could answer the following questions – in order to facilitate the understanding of this complex topic to readers: what potential does this study hold? What are the knowledge gaps and how do researchers tackle them? How do you see this area unfolding in the next 5 years? We think it would be extremely interesting for the readers.

Thank you for pointing it out. We have added a phrase to mention the potential of our study, knowledge gaps and futures in this area in the Discussion section (Line 291-296).

“Although the real consequences of the COVID-19 pandemic are yet to be known, we consider that measuring its impact on cancer care uncovers the stability, strengths and weaknesses of the healthcare system in a country. In addition to our results, understanding patients’ experiences during the pandemic and examining both short- and long-term cancer survival may provide insights into which point of care can be improved to achieve better cancer outcomes.”

However, we think the authors should be acknowledged for their work. In fact, they correctly addressed an important topic, the methods sound good and their discussion is well balanced.

One additional little flaw: the authors could better explain the limitations of their work, in the last part of the Discussion.

Thank you for your comment. We have added some phrases to better explain the limitations of our study in the Discussion section (Line 359-363, 369-371).

“Our results on the relative change in the number of operations were comparable to the estimates derived from the National Clinical Database encompassing all surgical procedures in Japan [45]; therefore, we assume our results are valid. Nonetheless, our results may not be generalisable to other countries or other areas in Japan. It is because the scale and timing of the pandemic, healthcare system, and the baseline characteristics of a population, including socioeconomic situations, health-seeking behaviours and screening uptakes, differ originally between and within countries.”

“The final limitation is that information on elective or emergency operation, socioeconomic status, comorbidities and measures of performance status has been not linked to these results. Identifying the affected populations and assessing the detailed situation is a key to proper reallocation of resources and aligning supply with patients’ needs. More detailed data on these characteristics that can be derived from DPC data are now available and will be able to reveal which populations were most affected by COVID-19.”

We believe this article is suitable for publication in the journal although major revisions are needed. The main strengths of this paper are that it addresses an interesting and very timely question and provides a clear answer, with some limitations.

We suggest a linguistic revision and the addition of some references for a matter of consistency. Moreover, the authors should better clarify some points.

Thank you for your comment. A professional British language editor proofread the manuscript. We have modified some points for clarification according to the professional editor’s comments.

Reviewer #2: The authors analyzed GI cancer diagnosis and treatment patterns in Osaka, Japan, comparing the COVID pandemic and the time before. Since the impact of the COVID pandemic is different according to the medical environment of each country, the data in this paper will help compare the situation of each country.

We would like to thank you for taking the time to review our manuscript. Thank you for your encouraging comment.

---

## [Decision Letter · Decision Letter 1]

7 Sep 2022

Diagnosis and treatment of digestive cancers during COVID-19 in Japan: A cancer registry-based study on the impact of COVID-19 on cancer care in Osaka (CanReCO)

PONE-D-22-19182R1

Dear Dr. Saito ; 

We’re pleased to inform you that your manuscript has been judged scientifically suitable for publication and will be formally accepted for publication once it meets all outstanding technical requirements.

Kind regards,

Ahmet Murt

Academic Editor

PLOS ONE

Additional Editor Comments (optional):

Reviewers' comments:

Reviewer's Responses to Questions

**Comments to the Author**

1. If the authors have adequately addressed your comments raised in a previous round of review and you feel that this manuscript is now acceptable for publication, you may indicate that here to bypass the “Comments to the Author” section, enter your conflict of interest statement in the “Confidential to Editor” section, and submit your "Accept" recommendation.

Reviewer #1: All comments have been addressed

Reviewer #2: All comments have been addressed

2. Is the manuscript technically sound, and do the data support the conclusions?

Reviewer #1: Yes

Reviewer #2: Yes

3. Has the statistical analysis been performed appropriately and rigorously? 

Reviewer #1: Yes

Reviewer #2: Yes

4. Have the authors made all data underlying the findings in their manuscript fully available?

Reviewer #1: Yes

Reviewer #2: Yes

5. Is the manuscript presented in an intelligible fashion and written in standard English?

Reviewer #1: Yes

Reviewer #2: Yes

6. Review Comments to the Author

Reviewer #1: The authors addressed all the queries and issues that the reviewers previously reported. We recommend Acceptance.

Reviewer #2: The authors have addressed most of my concerns in the revised version of the manuscript and improved it very much. I would think that it is now acceptable.

7. PLOS authors have the option to publish the peer review history of their article (what does this mean?). If published, this will include your full peer review and any attached files.

Reviewer #1: No

Reviewer #2: No

---

## [Editor Report · Acceptance letter]

11 Sep 2022

PONE-D-22-19182R1 

Diagnosis and Treatment of Digestive Cancers during COVID-19 in Japan: A Cancer Registry-based Study on the Impact of COVID-19 on Cancer Care in Osaka (CanReCO) 

Dear Dr. Kajiwara Saito:

I'm pleased to inform you that your manuscript has been deemed suitable for publication in PLOS ONE. Congratulations! Your manuscript is now with our production department. 

Kind regards, 

on behalf of

Dr. Ahmet Murt 

Academic Editor

PLOS ONE